# Study on the Effect of EZH2 Inhibitor Combined with TIGIT Monoclonal Antibody against Multiple Myeloma Cells

**DOI:** 10.3390/ijms24108603

**Published:** 2023-05-11

**Authors:** Zhaoyun Liu, Yue Jia, Chun Yang, Hui Liu, Hongli Shen, Hao Wang, Rong Fu

**Affiliations:** Department of Hematology, Tianjin Medical University General Hospital, 154 Anshan Street, Heping District, Tianjin 300052, China

**Keywords:** multiple myeloma, EZH2, NK cells, TIGIT

## Abstract

EZH2, a member of the polycomb repressive complex 2, induces trimethylation of the downstream gene at the histone three lysine 27 (H3K27me3) position to inhibit tumor cell proliferation. Here, we showed that the apoptosis rate and apoptotic protein expression increased after EZH2 inhibition, whereas key molecules of the NF-κB signaling pathway and the downstream target genes were inhibited. Additionally, the expression of CD155, a TIGIT high-affinity ligand in multiple myeloma (MM) cells, was decreased by the mTOR signaling pathway. Furthermore, the combination of EZH2 inhibitor and TIGIT monoclonal antibody blockade enhanced the anti-tumor effect of natural killer cells. In summary, the EZH2 inhibitor not only plays an anti-tumor role as an epigenetic drug, but also enhances the anti-tumor effect of the TIGIT monoclonal antibody by affecting the TIGIT-CD155 axis between NK cells and MM cells, thus providing new ideas and theoretical basis for the treatment of MM patients.

## 1. Introduction

Multiple myeloma (MM) is a plasma cell disease that produces monoclonal immunoglobulins (Ig) and is characterized by anemia, hypercalcemia, bone disease, and kidney damage. The prognosis for MM has improved significantly with the use of therapies such as proteasome inhibitors, immunomodulators, and autologous hematopoietic stem cell transplantation, but it is still incurable and there are persistent problems of relapse and drug resistance.

EZH2 is a methyltransferase that modifies the target gene promoter histone 3 at the Lys 27 site, which in most cases silences oncogene expression and affects activation or inhibition of key signaling pathways, further affecting several life processes such as cell proliferation, apoptosis, migration, and differentiation [1,2]. In hematologic malignancies, research studies and treatments targeting EZH2 are widely available. In MDS, EZH2 is often expressed as a loss-of-function mutation and leads to impaired hematopoietic differentiation [3]. In contrast, in myeloid/gonadal leukemia, EZH2 acts as a bifacial factor regulating cell fate [4]. Tazemetostat, the first FDA-approved epigenetic drug, has a significant effect on the human lymphoma cell line WSU-DLCL2, with mutations in the EZH2 SET domain site Y641F locus [5].

As the research progresses, the potential of EZH2 in MM is gaining attention. Previous studies have shown that aberrant and random multilocus methylation mutates myeloma cells to adapt to their environment [6]. However, EZH2, as a methyltransferase, demonstrates overexpression rather than mutation as the primary mechanism in MM, which exerts carcinogenesis and drug resistance, and is associated with poor prognosis [7,8,9]. A longitudinal study on MM demonstrated that the estimated median progression-free survival and median survival of patients with EZH2 overexpression were significantly lower than those without EZH2 overexpression [10]. A variety of small molecule inhibitors of EZH2 have been developed and submitted to experimental studies. On the one hand, inhibition of EZH2 can directly lead to an increased apoptosis of MM cells and exert anti-tumor effects [11]. On the other hand, the effect of an inhibitor is not limited to the tumor cells themselves; it still has critical impacts on other components of the immune microenvironment. Studies have revealed that EZH2 inhibition can enhance the recognition and killing function of immune cells [12]. EZH2 can target immune cells–for example, inhibition of EZH2 activates the quantity and quality of natural killer (NK) cells [13]. Side population cells (SP cells) are a group of cells with tumor stem cell characteristics, and the remaining tumor stem cells in the bone marrow after chemotherapy are the key to MM relapse. Dual inhibition of EZH1/2 targets SP cells, compensating for the deficiencies of conventional therapy [14].

In the meantime, the combination of EZH2 inhibitors with conventional drugs is coming into view. Although drugs such as immunomodulators and proteasome inhibitors have significantly improved the prognosis of MM, the problems of drug resistance and relapse persist, no matter how the sub-generation is updated. EZH2 has been identified as an initiating factor enriched in the genome of relapsed/refractory MM patients [15], and acquired drug resistance is also highly associated with genome-wide increased methylation and activation [16]. Multi-targeted combination therapy is the key to alleviating these problems. A combination of EZH2 inhibitors promotes the epigenetic reprogramming of tumor cells and reverses the process of resistance to immunomodulators [16,17,18]. Meanwhile, the combination of EZH2 inhibitors with proteasome inhibitors to induce tumor regression in vivo [12], and the feasibility of synthetic dual-target inhibitors, has been put on the agenda [19], but there is still insufficient evidence for clinical application. The above studies suggest that EZH2 not only has anti-tumor and immunomodulatory effects, but also has multifaceted therapeutic potential in combination with other classical drugs.

Several selective EZH2 inhibitors have been designed and investigated in pre-clinical trials and clinical trials, including GSK126, GSK343, UNC1999, E7438, EPZ005687, CPI-1205, etc. [20]. Studies on small molecule EZH2 inhibitors such as GSK126, UNC1999, and GSK343 have demonstrated their effectiveness in inducing apoptosis in MM cell lines [11,21]. Hernando et al., found that treatment of MM cell lines with the EZH2 inhibitor E7438 affected the proliferation and adhesion of tumor cells [22]. Another study using the EZH2 inhibitors EPZ005687 and UNC1999 in MM cell lines and patient samples confirmed that EZH2 inhibitors upregulate cell cycle control genes and reduce MM cell viability, leading to cell cycle arrest and apoptosis [18]. In addition, the EZH2 inhibitor GSK503 was proven to be effective in restoring RB transcriptional corepressor 1 transcription and suppressing tumorigenesis [23]. GSK503 inhibits the methyltransferase activity of wild-type and mutant EZH2 with high selectivity [24]. Here, we chose GSK503 with which to investigate the mechanism of an EZH2 inhibitor on MM apoptosis and to explore its potential application in combination therapy.

T cell immunoreceptor with Ig and ITIM domain (TIGIT) is an immune checkpoint receptor known to negatively regulate T cell function and is primarily involved in the immunosuppression of NK cell subsets and T cells, competing with the activating receptor CD226 to bind ligands. The relationship is similar to that of the combination of CTLA-4/CD28 receptors, which jointly maintain the balance of immune function. In the tumor microenvironment, however, this balance is tilted towards TIGIT, leading to tumor immune escape and immune depletion of effector cells, characterized by the upregulation of TIGIT and its multiple ligands [25]. The ligands of TIGIT include CD155, CD112, and CD113, of which CD155, also known as poliovirus receptor (PVR), has the highest affinity and has received widespread attention [26,27]. CD155 also serves as a potential prognostic marker in MM and is associated with disease staging and poor prognosis [28].

Targeting TIGIT has a bright future in MM, which is known for its altered immune environment, both in diagnosis and in treatment. Anti-TIGIT monoclonal antibodies entered clinical trials and the combination of anti-TIGIT monoclonal antibodies with anti-PD-1 monoclonal antibodies was expected to be effective. However, the results of monotherapy or combination therapy were not satisfactory [29]. This has forced researchers to explore more promising options for co-application. Is it possible to conceive of a solution in which a core pathway is chosen for a ‘multi-point attack’ under the premise of drugs containing different mechanisms?

Recently, NK cells and TIGIT have gradually entered the limelight [28]. In the bone marrow and peripheral blood of MM patients, based on the long-term impact of the tumor environment, NK cells showed immune exhaustion, characterized by upregulation of TIGIT and its multiple ligands. An anti-human TIGIT IgG 4-type monoclonal antibody, MG1131, significantly enhanced the anti-tumor activity of NK cells by competitively binding TIGIT [30]. However, the current research on TIGIT is still mostly focused on T cells, while less attention has been paid to NK cells, and the mechanism is still unclear. Hence, this study focused on NK cells in order to explore the mechanism of action initially.

Based on this, this study identified a therapeutic strategy consisting of a combination of small molecule methyltransferase inhibitors and blocking monoclonal antibodies that target both ligands and receptors of the CD155-TIGIT axis, significantly increasing the anti-tumor effect of NK cells. In this study, EZH2 inhibitors not only exerted anti-tumor effects as epigenetic agents, but also enhanced the anti-tumor effects of TIGIT monoclonal antibodies by affecting the TIGIT-CD155 axis between NK cells and MM cells, providing new ideas and preliminary theoretical basis for the treatment of MM patients.

## 2. Results

### 2.1. EZH2 Inhibitor GSK503 Enhanced the Apoptosis of MM Cell Line in a Concentration-Dependent Manner

Flow cytometry was used to detect the apoptosis levels in the control and experimental groups under different concentrations of GSK503 treatment. We observed that the apoptosis of MM cell lines gradually increased as the working concentration of GSK503 increased (Figure 1A,B, both *p* < 0.05), and the cells were almost all apoptotic at 30 µM. (RPMI-8226: 5.21 ± 0.19% (Ctrl), 7.05 ± 0.11% (5 µM), 8.63 ± 0.37% (10 µM), 26.73 ± 8.73% (15 µM), and 97.70 ± 0.62% (30 µM); OPM-2: 2.08 ± 0.68% (Ctrl), 5.36 ± 0.73% (5 µM), 26.56 ± 4.21% (10 µM), 68.91 ± 2.52% (15 µM), and 95.90 ± 1.81% (30 µM)).

Thereafter, we used Western blot to verify that the EZH2 inhibitor GSK503 inhibited the downstream functional protein H3K27me3 of EZH2 in MM cell lines OPM-2 and RPMI-8226 in 10 µM (Figure 1C). Subsequently, Western blot showed that GSK503 increased the expression of pro-apoptosis-related proteins BAX and cleaved caspase-3, while the expression of anti-apoptosis-related protein BCL-2 was decreased (Figure 1D). The above results confirmed the concentration-dependent enhancement of apoptosis levels in MM cell lines by GSK503 and helped us to select the 10 µM concentration for subsequent experiments.

### 2.2. GSK503 Inhibits the NF-κB Signaling Pathway

Western blot assessment of the phosphorylation levels of key proteins in the NF-κB signaling pathway showed that, in the presence of GSK503, the NF-κB pathway was inhibited, with a decrease in p-p65 and p-STAT3 expression, accompanied by an increase in IκBα (Figure 1E). At the same time, the expression levels of IL-6 and Bcl-2, the downstream target genes of this pathway, significantly decreased (Figure 1F). The above results imply that the pro-apoptotic effect of EZH2 on MM cells may be related to the NF-κB signaling pathway.

### 2.3. GSK503 Reduced the Expression of TIGIT Ligand CD155 on the Surface of MM Cell Line

OPM-2 cells with high CD155 expression were selected for subsequent experiments. The relative RNA expression level of the TIGIT ligand CD155 on the surface of MM cells was significantly reduced after GSK503 treatment (Figure 2A). The expressions of CD155, CD112, and CD113 on the surface of MM cells were detected by flow cytometry. CD155 expression was significantly reduced (Figure 2B,C). The expression of CD155 on the surface of OPM-2 cells after GSK503 working concentration, from high to low, was 73.51 ± 1.19%, 68.77 ± 2.67%, 59.72 ± 1.38%, 25.95 ± 1.32%, and 27.60 ± 4.50%. CD112 and CD113 did not change significantly, or were expressed at lower levels and lacked comparative significance (Figure 2D).

### 2.4. EZH2 Inhibitor GSK503 Regulates MM Cell Surface CD155 Expression via the mTOR Signaling Pathway

We performed RNA sequencing on OPM-2 cells after 72 h of GSK503 (10 nM). According to GO (Gene Ontology) analysis (Figure 2E), the differences in biological processes are mainly reflected in cellular processes and biological regulation: at the cell component level, the difference is mainly reflected in cell part; in the molecular function, it is mainly reflected in protein binding. The KEGG biological pathway enrichment analysis showed that the differential genes were more enriched in the mTOR signaling pathway (Figure 2F).

Western blot results confirmed that the phosphorylated mTOR levels of MM cells significantly decreased after 72 h of GSK503 treatment (Figure 2G,H, *p* < 0.05). Subsequently, we used the mTOR signaling pathway inhibitor rapamycin to act on OPM-2 cells and found that it reduced cell surface CD155 expression in a concentration-dependent manner. (Figure 2I,J: The expression of CD155 on the surface of OPM-2 cells after rapamycin working concentration, from high to low, was 36.45 ± 1.64%, 31.33 ± 0.51%, 20.95 ± 1.52%, 15.91 ± 0.84, and 15.02 ± 0.82%).

These results demonstrate that EZH2 inhibitor GSK503 regulates MM cell surface CD155 expression via the mTOR signaling pathway.

### 2.5. EZH2 Inhibitor GSK503 Enhances the Anti-Tumor Effect of TIGIT Monoclonal Antibody and NK Cell Function

Since EZH2 inhibition was observed to have a significant inhibitory effect on the expression of CD155, we speculated that this phenomenon may have an additional effect on the anti-MM cell process of NK cells involved in the TIGIT-blocking monoclonal antibody. In order to test this conjecture, we performed in vitro co-culture experiments with NK-92 cells and OPM-2 cells, divided into four groups according to whether they were pretreated with GSK503 and whether the TIGIT-blocking monoclonal antibody was added (Figure 3A). The results confirmed that pretreatment with GSK503 only (28.01 ± 12.29%), or addition of the TIGIT-blocking monoclonal antibody only (28.47 ± 8.07%), increased the anti-tumor effect of NK cells compared to the control group (14.15 ± 8.30%), while the dual treatment group (40.11 ± 10.36%) showed the best effect (Figure 3C–E).

Subsequently, mononuclear cells were extracted from the bone marrow of NDMM patients. The cells were grouped according to the graph (Figure 3B), and CD138+ MM cells were labeled by flow cytometry (Figure 3F). The killing effect of NK cells on CD138+ MM cells and the change of NK cell function in the system were observed by flow cytometry. The results confirmed that the killing function of NK was significantly enhanced in the GSK503-treated group (34.32 ± 16.15%) and the TIGIT-blocking monoclonal antibody group (28.31 ± 16.98%) compared with the control group (25.29 ± 16.07%), and the combined application group (44.94 ± 17.19%) killed the most (Figure 3G,H). On the other hand, NK cell function was also partially restored, mainly reflected by the increase in NK cell surface functional molecule NKP30 in the GSK503-treated group (48.79 ± 22.28%) and the TIGIT-blocking monoclonal antibody group (40.49 ± 15.78%) compared with the control group (33.70 ± 14.49%), and in the system with both GSK503 and TIGIT-blocking monoclonal antibody (60.99 ± 23.32%), the above effect was the most pronounced (Figure 3I–K).

The above findings confirm that the TIGIT-blocking monoclonal antibody can significantly enhance the function and anti-tumor effect of NK in the MM tumor microenvironment compared to EZH2 inhibition alone or TIGIT blockade in the presence of EZH2 inhibition.

## 3. Discussion

MM is the second most common hematological malignancy, generating monoclonal plasma cells associated with immunoglobulin production. Previous studies have shown that the expression of EZH2 plays an essential role in the developmental differentiation of B cells [31]. In MM, EZH2 overexpression is involved in biological processes including proliferation, apoptosis and side-population maintenance of tumor cells, and is also associated with cellular drug resistance. Previous studies have demonstrated that EZH2 inhibitors regulate the NF-κB p65 pathway and influence anti-inflammatory and anti-tumor processes [32,33], but the relevance of EZH2 to the NF-κB pathway has not been investigated in MM. In our study, we have demonstrated that inhibition of EZH2 promotes apoptosis in MM cells and is associated with the NF-κB pathway. In addition, EZH2 expression promotes the immune escape of tumor cells. In Waldenstrom macroglobulinemia, EZH2 promotes the expression of pro-apoptotic genes, reduces the sensitivity of tumor cells to external toxicity-inducing signals, and escapes NK cell-mediated death, enabling immune escape, which makes EZH2 a potential target for chemosensitization [34]. Our results also indicated that the expression of pro-apoptotic proteins BAX and cleaved caspase-3 was increased, while the expression of anti-apoptotic protein Bcl-2 was decreased under the effect of EZH2 inhibitors. However, the underlying mechanisms of how EZH2 affects apoptotic proteins and causes immune escape need to be further explored in subsequent experiments. Notably, the role of EZH2 is not limited to tumor and immune cells, but affects the entire tumor microenvironment. For example, EZH2 promotes tumor angiogenesis by silencing vasohibin1 in vascular endothelial cells [35], which suggests that we need to evaluate the advantages and disadvantages of EZH2 application in a holistic manner, and investigate whether other drugs need to be used in combination to avoid shortcomings when applied clinically.

In recent years, the targeting of immune checkpoints has received much attention. However, conventional immune checkpoint antibodies have disadvantages, such as high cost, poor penetration, and autoimmune side effects, and the outcome of their individual application in clinical trials has not reached expectations—all factors which have limited their clinical application. As a result, the therapeutic strategy of small molecule inhibitors combined with an immune checkpoint monoclonal antibody has gradually attracted attention. Previous studies have found that in high-grade plasmacytotic ovarian cancer, focal adhesion kinase (FAK) small molecule inhibitors reduce tumor load, inhibit CD155, CD112, and PD-L1 levels on KMF (Kras, Myc, FAK) cells, and prolong host survival when combined with TIGIT-blocking monoclonal antibodies [36]; in breast cancer, the combination of small molecule inhibitors targeting DNA methyltransferases and TIGIT monoclonal antibodies was found to better inhibit tumor metastatic potential [37]. In addition, some small molecule inhibitors with similar anti-immune checkpoint effects are also attractive, most of which target the PD-1/PD-L1 axis, including PDI-1 [38], the MMP2/9 inhibitor SB-3CT [39] and BMS-202 nanoparticles [40]. In the field of EZH2-related studies, it has been suggested that EZH2 is closely related to the expression of the immune checkpoint ligand PD-L1 [41,42], whereas studies targeting other immune checkpoints are scarce.

Generally, EZH2, as a classic oncogene inhibitor, is characterized by a direct anti-tumor effect. However, our research demonstrated that EZH2 also plays a role in targeting and regulating CD155, the high affinity receptor for TIGIT. Our previous studies proved that the TIGTI-CD155 axis between NK cells and MM cells is a prospective immunotherapeutic target in the bone marrow microenvironment of MM [43]. However, due to the high heterogeneity of CD155 expression in myeloma cells, this receptor–ligand interaction in NK cells and myeloma cells remains to be further clarified. Therefore, in this study of EZH2 inhibitors involved in MM treatment, we believe that the relationship between EZH2 and CD155 deserves to be taken into account, and attempted to combine treatment with a TIGIT monoclonal antibody, with the aim of achieving a dual inhibition of the TIGIT-CD155 axis. In this study, we determined the effect of EZH2 small molecule inhibitors on the expression of TIGIT ligands and found that the expression of CD155 was significantly inhibited in a concentration-dependent manner. While the inhibition of such ligands is usually dependent on immune checkpoint monoclonal antibodies or inhibitors, EZH2 small molecule inhibitors are also synergistic and can still promote the apoptosis of tumor cells themselves. Subsequently, mRNA sequencing, GO analysis, and KEGG analysis were performed for the control and experimental groups in order to clarify the signaling pathway mechanism underlying the effect of EZH2 inhibitors in reducing CD155 expression. The sequencing results showed that the differential gene was enriched in the mTOR signaling pathway. Based on the RNA sequencing results, we further investigated the important role of the mTOR signaling pathway in the inhibition of CD155 by EZH2 inhibitors. Western blotting confirmed that the level of phosphorylated mTOR in MM cells significantly decreased after EZH2 inhibitor treatment. FCM showed that the expression of CD155 on the surface of MM cells also decreased in a concentration-dependent manner after the EZH2 inhibitor and mTOR pathway inhibitor rapamycin. It is suggested that EZH2 is involved in the expression of CD155 on the surface of MM cells through the mTOR signaling pathway.

The key issue we also need to consider is that TIGIT binds to CD112 and CD113 ligands, in addition to CD155 ligands, on the surface of myeloma cells, but our experiments did not observe significant changes in the expression of the remaining ligands after EZH2 inhibitor treatment [26]. In addition, our previous experiments confirmed that CD155 is also expressed on the surface of marrow mesenchymal stem cells [43]. These all limit the efficacy of dual targeting of the TIGIT-CD155 axis between MM cells and NK cells when EZH2 is used in combination with a TIGIT monoclonal antibody.

We then performed in vitro co-culture experiments with NK-92 and MM cell lines/BM CD138+ cells from NDMM patients. Additionally, we demonstrated that the anti-tumor effect of NK cells was significantly enhanced by EZH2 inhibitors, and when EZH2 small molecule inhibitors were combined with TIGIT-blocking monoclonal antibodies, the anti-tumor effect of NK cells was higher than that of cells receiving monotherapy, and the expression of NKP30, an NK immune molecule, was increased in primary cells. These results are consistent with previous studies and further explore the underlying mechanisms. This addition would be very beneficial due to the glowing promise of therapies targeting immune checkpoints in MM.

## 4. Method

### 4.1. Patients

A total of 5 newly diagnosed multiple myeloma (NDMM) patients admitted to the Hematology Department of Tianjin Medical University General Hospital were included in this study. The study protocol was approved by the Ethics Committee of Tianjin Medical University General Hospital (ethics committee code NO.IRB2015-YX-009, September 2022) and was conducted in accordance with the Declaration of Helsinki, and all subjects provided signed informed consent.

### 4.2. BMMNCs

Bone marrow was collected from the iliac bones of NDMM patients, and bone marrow mononuclear cells (BMMNCs) cells were isolated using density gradient centrifugation (Solarbio, Beijing, China) and cultured in RPMI-1640 medium (Solarbio, Beijing, China) supplemented with 20% fetal bovine serum (FBS, Gibco, Waltham, MA, USA), 200 U/mL recombinant IL-2 (Procell Life Science & Technology Co., Ltd., Wuhan, China), 100 U/mL penicillin, and 1% chain penicillin (Gibco, Waltham, MA, USA) at 37 °C with 5% CO_2_. The cultures were inoculated in 24-well plates at a density of 1 × 10^6^/mL and incubated for 3 days for subsequent experiments.

### 4.3. Cells and Cell Culture

Human myeloma cell line OPM-2 (Peking Union Cell Bank, Beijing, China) was cultured in RPMI-1640 medium (Solarbio, Beijing, China) supplemented with 10% fetal bovine serum (FBS, Gibco, Waltham, MA, USA) and 1% chain penicillin (Gibco, Waltham, MA, USA) at 37 °C with 5% CO_2_. The cell lines were inoculated at a density of 1 × 10^6^/mL in T25 flasks and passaged for subsequent use when the density reached 90%.

Human myeloma cell line RPMI-8226 (Guangzhou Cellcook Biotech Co., Ltd., Guangzhou, China) was cultured at 37 °C with 5% CO_2_ using DMEM basic medium (Gibco, Waltham, MA, USA) supplemented with 20% fetal bovine serum and 1% chain penicillin (Gibco, Waltham, MA, USA). Cell lines were inoculated at a density of 1 × 10^6^/mL in T25 culture flasks and passaged for subsequent use when the density reached 90%.

NK cell line NK-92 was cultured using NK-92 complete medium with 200 U/mL and the recombinant IL-2 (All from Procell Life Science & Technology Co., Ltd., Wuhan, China) was incubated at 37 °C with 5% CO_2_. The cell lines were inoculated in T25 culture flasks at a density of 2–3 × 10^6^/mL, using alternating half volume fluid changes and centrifugal fluid changes, and passaged for backup according to cell growth.

### 4.4. Western Blot

Western blot was used to assess the levels of proteins in cell extracts after the addition of GSK503 (10 µM). Cells were lysed using RIPA buffer (Sigma-Aldrich, St. Louis, MO, USA). Extracted protein concentrations were determined using a BCA Protein Assay Kit (Sigma-Aldrich, St. Louis, MO, USA). Whole cell lysates were subjected to sodium dodecyl sulfate–polyacrylamide gel electrophoresis (SDS-PAGE). After SDS-PAGE, the samples were transferred to a polyvinylidene difluoride (PVDF) membrane (Easybio, Beijing, China). After antibody incubation and washing, an ECL chemiluminescence kit (Cell Signaling Technology, Danvers, MA, USA) was used and exposed.

### 4.5. RNA Isolation and Real-Time Fluorescent Quantitative PCR (RT-PCR)

RT-PCR was used for relative expression of target gene mRNA after treatment of the MM cell line with EZH2 inhibitor GSK503 (10 µM). TRIzol, chloroform, isoamyl alcohol, and ice ethanol (Tianjin Chemical Reagent Factory) were used to extract RNA. Complementary DNA was obtained using a fast reverse transcription kit (Tiangen Biotech, Beijing, China). SYBR fluorescence (Bimake, Houston, TX, USA) was used for amplification reaction in 20 μL system.

### 4.6. RNA Seq Analysis

OPM-2 was divided into two groups, with or without treatment with the EZH2 inhibitor GSK503 (10 μM), and cells were collected after 72 h and stored in 1 mL Trizol (Thermo Fisher Scientific, Waltham, MA, USA). Total RNA sequencing and bioinformatics analysis were completed by Wuhan Kangshi Technology Co., Ltd. (Wuhan, China).

### 4.7. Co-Culture MM Cells and NK-92 Cells

MM cell line OPM-2 and bone marrow (BM) CD138+ cells from NDMM patients were treated with 10 μΜ GSK503 (Selleck, Shanghai, China). GSK503 inhibitor powder was dissolved in DMSO to a concentration of 10 mM, dispensed and stored in a −80 °C refrigerator. The densities of OPM-2 cells and BM CD138+ cells were adjusted to 1 × 10^6^ cells/mL. Subsequently, 1 μL of GSK503 storage solution was added to each 1 mL of cell suspension to make a working concentration of 10μM. After 3 days of incubation, the cells were removed, centrifuged, and resuspended and reinoculated in 24-well plates, so that the inoculum volume was controlled at 5 × 10^5^/well. OPM-2 cells were then stained with calcein and washed.

A co-culture system of OPM-2 cells/BM CD138+ cells with fluorescence was established with NK-92 cells in a 1:1 ratio and divided into four groups: blank control group, EZH2 inhibitor alone group, TIGIT monoclonal antibody alone group, and combined EZH2 inhibitor and TIGIT monoclonal antibody group. According to the experimental requirements, 10 µM of TIGIT monoclonal antibody (Thermo Fisher Scientific) was added to the appropriate groups. Each group was placed in a cell culture incubator for 4 h, and then used for subsequent experiments.

### 4.8. Annexin V/7-AAD Apoptosis Assay Kit to Detect Apoptosis

MM cells were collected after treatment with different concentrations of GSK503 and washed twice with PBS (1500 rpm × 5 min), collected in groups of 1~5 × 10^5^ cells, and resuspended with 100 μL 1 × Binding Buffer. Subsequently, 5 μL 7-AAD staining solution and 5 μL Annexin V-PE staining solution were added and shaken to mix well. The cells were incubated for 15 min at room temperature and detected on a Beckman Coulter flow cytometer (Beckman Coulter Company, Brea, CA, USA; Purchased from Shanghai, China) within one hour.

Cells were collected from the OPM-2 and NK cell co-culture system and stained with the appropriate reagents according to the above procedure. FITC green fluorescence was used to detect calcein-labeled OPM-2 cells. Annexin V was detected using the PE fluorescence channel, and 7-AAD was detected using the Per-cp channel. Beckman Coulter flow cytometry was used to detect the fluorescence in one hour.

Cells were collected from the BM CD138+ and NK cell co-culture system. CD138-APC labeling of CD138+ MM cells in the co-culture system was conducted. ECD and FITC staining solutions were added and shaken to mix well. The cells were incubated for 15 min at room temperature and detected on a Beckman Coulter flow cytometer within one hour.

### 4.9. Flow Cytometry for NK-92 Cell Function

The cells were collected from the co-culture system, washed twice with PBS (1500 rpm × 5 min), and collected in groups of 1~5 × 10^5^ cells. Subsequently, 100 μL PBS was added to resuspend the cells, and 5 μL each of CD3-Percp and CD56-APC staining solution were added and the mixture was incubated for 15 min at room temperature away from light. After two washes of PBS (1500 rpm × 5 min) to remove the residual stain, 100 μL of PBS was added again to resuspend the cells, and 5 μL NKP30-PB450 stain was added. The cells were washed 2 times with PBS (1500 rpm × 5 min) to remove residual staining solution. After circling NK-92 cells with CD3-CD56+ according to the above channels, NK cell functional molecule expression was detected using a Beckman Coulter flow cytometer, and the results were analyzed by Cyt-Expert (Beckman, Brea, CA, USA).

### 4.10. Statistics and Quantification

All experimental data were first tested for normal distribution before analysis. Data that conformed to a normal distribution were expressed as mean ± standard deviation. For two mutually independent random samples that conformed to a normal distribution, the independent samples *t*-test was used. For more than two mutually independent random samples that obey a normal distribution and satisfy chi-square, a one-way analysis of variance (ANOVA test/F-test) was used. For data from paired samples, a paired t-test was used. GraphPad Prism (version 8.0) statistical software was used for statistical analysis. The data were obtained through three repeated independent experiments. (*p* value < 0.05 was considered significant. * *p* < 0.05; ** *p* ≤ 0.01, *** *p* ≤ 0.001, **** *p* ≤ 0.0001).

## 5. Conclusions

In conclusion, we found that EZH2 inhibitors promote apoptosis in MM cell lines through the NF-κB signaling pathway and reduce TIGIT ligand CD155 expression on MM cell surfaces by inhibiting the mTOR signaling pathway. In addition, EZH2 inhibition enhanced NK cell anti-tumor effects and immune recovery when combined with the TIGIT-blocking monoclonal antibody (Figure 4). The dual inhibition of inhibiting an immune checkpoint and its ligand by small molecular antibodies combined with monoclonal antibodies opens a new window for clinical therapy strategies.

## Figures and Tables

**Figure 1 ijms-24-08603-f001:**
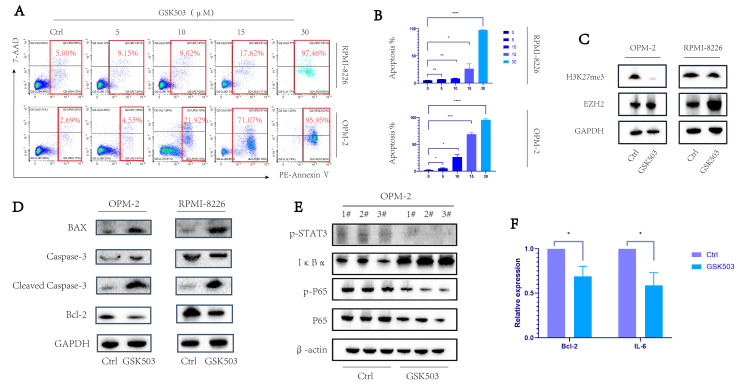
GSK503 concentration-dependently enhances apoptosis in MM cell by inhibiting the NF-κB signaling pathway. (**A**) MM cell apoptosis by flow cytometry after different concentrations of GSK503 treatment for 72 h. (**B**) When the concentrations of GSK503 (µM) were 0, 5, 10, 15, and 30, the apoptosis increased gradually after 72 h. (**C**) Treatment of OPM-2 and RPMI-8226 cell lines with 10 µM GSK503 for 72 h. GSK503 reduced the expression of H3K27me3, the functional protein downstream of EZH2. (**D**) The changes in the apoptosis-related protein cell lines BAX, caspase-3, cleaved caspase-3, and BCL-2 of OPM-2 and RPMI-8226 after 10 µL GSK503 treatment for 72 h. (**E**) In the OPM-2 cell line, WB demonstrated that, after 10 µM GSK503 treatment for 72 h, the expressions of p65, pp65, and pSTAT3 decreased, accompanied by the increase in IκBα expression, (**F**) and qRT-PCR showed that the expression of NF-κB downstream target genes Bcl-2 and IL-6 decreased. (**B**,**F**) used paired *t*-tests. *p* value < 0.05 was considered significant. Appendix A shows the gating strategy for flow cytometry. The results are presented as expression percentage and the experimental data are shown in Appendix A. * *p* < 0.05; ** *p* ≤ 0.01, *** *p* ≤ 0.001, **** *p* ≤ 0.0001.

**Figure 2 ijms-24-08603-f002:**
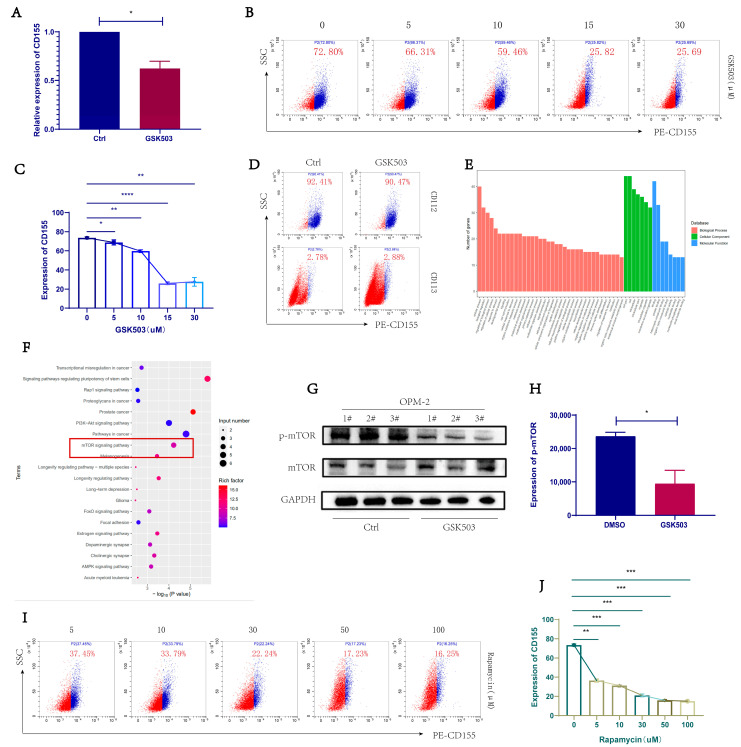
GSK503 inhibits TIGIT ligand CD155 expression on OPM-2 cell surfaces by inhibiting mTOR signaling pathway. OPM-2 cell line treated with 10 µL GSK503: (**A**) qRT-PCR showed that the relative expression of CD155 decreased, treated with GSK503 (10 uM) after 72 h. (**B**) CD155 expression by flow cytometry, treated with GSK503 for 72 h. (**C**) CD155 expression gradually decreased with increasing concentration of GSK503 in a concentration-dependent manner. (**D**) Expression changes of TIGIT ligands CD112 and CD113 on the surface of OPM-2 cells treated with GSK503(10 uM) after 72 h. OPM-2 cells were treated with GSK503 (10 uM) for 72 h and expression changes of TIGIT ligands CD112 and CD113 on the surface of OPM-2 cells were detected by flow cytometry. (**E**) RNA sequencing GO analysis. (**F**) RNA sequencing KEGG analysis. The red rectangle refer to the mTOR signaling pathway that is chosen as the object of study. (**G**,**H**) After 10 µL GSK503 treatment of OPM-2 cells for 72 h, (**H**) Western blot and (I) grayscale analysis detected phosphorylated mTOR protein expression. (**I**) CD155 expression by flow cytometry, treated with Rapamycin for 72 h. (**J**) CD155 expression gradually decreased with increasing Rapamycin concentration in a concentration-dependent manner. (**B**,**D**,**I**) are representative flow cytometry plot results from three independent replicate experiments. Appendix A shows the gating strategy for flow cytometry. The results are presented as expression percentage and the experimental data are shown in Appendix A. (**A**,**C**,**H**,**J**) used paired *t*-tests. *p* value < 0.05 was considered significant. * *p* < 0.05; ** *p* ≤ 0.01, *** *p* ≤ 0.001, **** *p* ≤ 0.0001.

**Figure 3 ijms-24-08603-f003:**
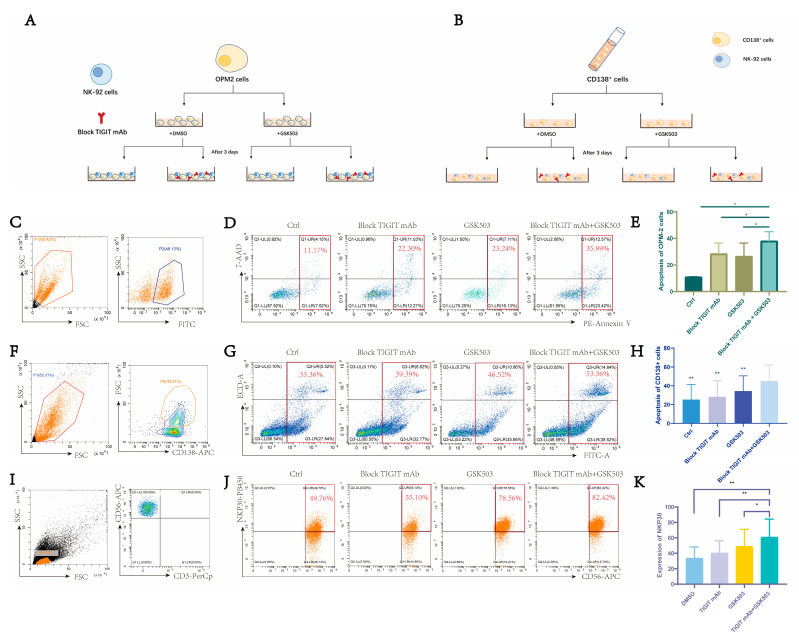
EZH2 inhibitor combined with TIGIT-blocking monoclonal antibody enhanced NK cell-induced MM cells apoptosis. (**A**) Flow diagram of OPM-2 co-culture with NK-92 cell line. (**B**) Flow diagram of NDMM bone marrow mononuclear cell processing. (**C**) OPM-2 cells were pre-stained using calcein, and then labeled as target cells on flow cytometry. (**D**) Apoptosis of OPM-2 cells under different treatment conditions after co-culture. (**E**) After 4 h of cell line co-culture, apoptosis of OPM-2 cells increased in the GSK503 pretreatment only group and in the group for which TIGIT-blocking monoclonal antibody was added only at co-culture, while the combination group demonstrated the most apoptosis. (**F**) CD138 + MM cells were labeled on flow cytometry. (**G**) CD138+ MM cell apoptosis in each group. (**H**) CD138+ MM cell apoptosis increased in the GSK503 and TIGIT-blocking monoclonal antibody groups, and the combination group showed the most significant increase. (**I**) CD3-CD56+ NK cells were labeled on flow cytometry. (**J**) CD3-CD56+ NK cells were labeled on flow cytometry. (**K**) The expression of NKP30 on the surface of primary NK cells was increased by GSK503 or TIGIT-blocking monoclonal antibody, and the combination group showed the highest increase. (**C**,**D**,**F**,**G**,**I**,**J**) are representative flow cytometry plots. Appendix A shows the gating strategy for flow cytometry. The results are presented as expression percentage and the experimental data are shown in Appendix A. (**E**,**H**,**K**) used paired *t*-tests. *p* value < 0.05 was considered significant. * *p* < 0.05; ** *p* ≤ 0.01.

**Figure 4 ijms-24-08603-f004:**
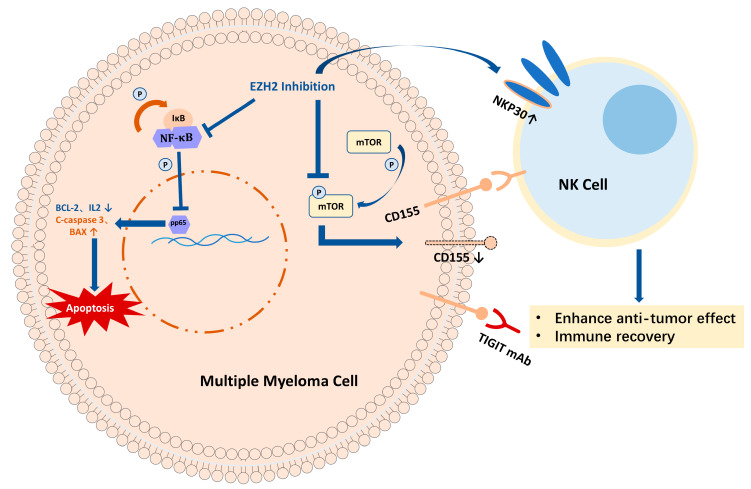
Summary of the role of EZH2 inhibitor combined with TIGIT-blocking monoclonal antibody in MM (Image partly from Servier Medical Art).

## Data Availability

All datasets that the conclusions of the paper rely on are available to readers and deposited in publicly available repositories.

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
