# Peer review of "Study on the Effect of EZH2 Inhibitor Combined with TIGIT Monoclonal Antibody against Multiple Myeloma Cells"

_ijms, 2023, doi:10.3390/ijms24108603_

Round 1

Reviewer 1 Report

The authors describe pathways affected by treatment identified as GSK503 for Multiple myeloma and its plausible mechanism for apoptotic activity.

They combine it with TIGIT monoclonal antibody treatment as a strategy to enhance the anti-tumor effects in patient derived MM cells as in vitro models. 

The study is relevant for understanding the potential benefits of a combination therapy. 

There are few comments to improve the understanding of the concepts presented in the manuscript.

 1. As GSK503 already known inhibitor of EZH2, it needs to be clearly stated and additional explanation for the selection of the inhibitor needs to be provided. 

 2. As written in Line no 221, measurement of the  IL6 is shown or (Fig 1F) is it IL2?  explain

 3. Fig 2F and 2G legends are not clear and also other figures would benefit from improvement of quality.

Author Response

Comment 1: As GSK503 already known inhibitor of EZH2, it needs to be clearly stated and additional explanation for the selection of the inhibitor needs to be provided.

Response: We thank the reviewer for pointing this out. As suggested by the reviewer, we have added to the manuscript the current use of EZH2 inhibitors in MM studies and supplemented the relevant literature to explain why we have chosen GSK503, an EZH2 inhibitor. (Line 78-91)

Comment 2: As written in Line no 221, measurement of the IL6 is shown or (Fig 1F) is it IL2? explain

Response: We apologize for the carelessness with which we have incorrectly marked the manuscript. Figure 1F shows the expression levels of IL-6 and Bcl-2, the target genes downstream of the NF-κB pathway, rather than IL-2 as we originally labelled. We have corrected the errors in the manuscript and highlighted them in red. (Line 259, 276)

Comment 3: Fig 2F and 2G legends are not clear and also other figures would benefit from improvement of quality.

Response: We are apologizing for the lack of clarity in all the images in our previous uploads. We have modified the legends and changed the format of all the images to vector images so that they are still sharp enough when enlarged.

Reviewer 2 Report

The manuscript describes the effect of EZH2 inhibitor and anti-TIGIT on NK cell activity against MM cell lines.  The idea of the manuscript is interesting, however, there are several major issues:
1) Authors should perform additional experiments as authors did not show the effect of inhibitors on NK cell enhancement against  MM. Apoptosis assay is not correct assay to be used for demonstrating NK cell cytotoxicity.  Authors should perform NK cell cytotoxicity assays to demonstrate the effect of blockade on NK cell enhancement against MM. 
2) Figure 2B is unclear. Authors presented raw flow cytometry plots. There was no difference between 15 and 30uM. Figures 2C and 2D are graph versions of the Figure 2A and 2B ? If so, authors should clearly state that under the Figure that 2A and 2B are representative  flow cytometry plots. Similar should be done for flow cytometry results presented in the  Figure 3. It is also unclear whether authors used cell viability dye for the Figure 2A-2D and Figure 3? What does «expression of » mean for the graphs 2C and 2D? Is it MFI? If so, please state. Furthermore, why the authors compared [0]concentration of GSK503 and [5uM ]of Rapamycin in the Figure 2B? 
3) Authors should provide gating strategy for all flow cytometry results in the Supplementary material 

4) Figures 3J and 3 K are very confusing. There is no statistical significance between DMSO and GSK503 but there is a statistical significance between GSK503 and anti-TIGIT+GSK503, though there is only 3.66% difference between the groups? Authors should present all flow cyrometry data and the expression level using MFI.

5) in order to demonstrate NK cell functional activity, authors must perform degranulation assay and IFN gamma production by NK cells in response to GSK503 and anti-TIGIT in comparison with appropriate controls.

6) All concentration  should be clearly stated under all Figures 

Additional major  comments:

1) All Figures are hard to read. All flow cyrometry plots are impossible to read. These Figures are  not acceptable for publication 

2) Materials and method section does not include all necessary information  including the information regarding bioinformatics, concentration of inhibitors, etc

3) Abstract should be re-written. In the present form abstract is not acceptable for publication. 

Reviewer 3 Report

The authors of the manuscript titled "Study on the effect of EZH2 inhibitor combined with TIGIT monoclonal antibody on promoting NK cells against multiple myeloma cells" have presented an interesting story. However, due to very poor presentation of figures, many of which are unreadable, insufficient information about the axes, scale bar on FACS plots, and insufficient information on the statistical tests performed, it is rather difficult to review the manuscript.

Figures are not readable in printed or very much zoomed in version of the pdf, making the conclusions made difficult to judge (many sub-figures in each figure). I recommend the authors correct for these figure quality issues and resubmit the manuscript.

In the bar graphs for FACS analysis data performed in figure 3, the error bars are overlapping and still have statistically significant differences. It is therefore essential that the authors present the individual data points in their whole manuscript and also clearly mention the statistical test performed.

I do not see the manuscript fit for publication and recommend the authors to resubmit after revising for the above points, since the manuscript could not be reviewed in the current figure quality.

Author Response

Comment 1: Figures are not readable in printed or very much zoomed in version of the pdf, making the conclusions made difficult to judge (many sub-figures in each figure). I recommend the authors correct for these figure quality issues and resubmit the manuscript.

Response: We appreciate the reviewer's advice. We have changed the format of all images to vector and re-uploaded them so that they remain sharp enough when enlarged.

Comment 2: In the bar graphs for FACS analysis data performed in figure 3, the error bars are overlapping and still have statistically significant differences. It is therefore essential that the authors present the individual data points in their whole manuscript and also clearly mention the statistical test performed.

Response: In Figure 3, we compared the control and monotherapy groups to the combination group in MM cell line and NDMM patients (Line 338-352) and labelled statistically significant comparisons in the figures. In our manuscript we did not make comparisons between other groups. We have marked the statistical methods used in the manuscript in accordance with the reviewer's comment and highlighted in red. The revised manuscript has been uploaded and we are grateful to the reviewer for the attention and valuable comments on this manuscript.

Round 2

Reviewer 2 Report

Authors improved the manuscript, however, authors did not address all the comments and there are still a lot of major concerns:

  1. In the title it is stated that EZH2 inhibitor combined with anti-TIGIT enhance NK cell activity, however, the authors did not perform any of NK cell functional/cytotoxic  assays to prove that. Authors only measured NKp30 expression and performed apoptosis assay. Since authors did not include any assay that proves NK cell enhancement, authors must change the title accordingly. 
  2. Authors did not provide gating strategy for any flow cytometry results. Authors should provide not a written gating strategy but a gating strategy in the Figure formats
  3. There was no cell viability dye for many of the experiments ( including for the Figure 2B/2I, 2D, Figure 3 I/G) which means that a lot of effects seen may be due to cell death and therefore unspecific fluorescence. It is not correct to perform flow cytometry experiments without cell viability dye as is a part of gating strategy. 
  4. Authors incorrectly presented representative flow cytometry plots. Even though authors wrote in red colour percentage of positive cells, however, all flow cytometry figures are unreadable. Authors should have a look on how researchers present representative flow cytometry plots and flow cytometry data. All information in all Figures must be readable. 
  5. Figures 3D, G and J are incorrect. Authors highlighted double positive cells in red gating and single positive cells, however in red colour authors presented percentage for one type of cells. For example, in the Figure 3D, there is overall red gate for AAD+ Annexin V+ cells as well as for Annexin V+ cells, however the percentage is 17.68%. Percentage should be for each cell population  separately. Furthermore, information in representative flow cytometry gates should be readable 
  6. Figures 2C and 2D (revised to Figure 2C and Figure 2J) are NOT MFI. These figures show percentage and not MFI. Please refer again to the 1st round of comments and revise accordingly 

Reviewer 3 Report

The authors of the manuscript titled "Study on the effect of EZH2 inhibitor combined with TIGIT monoclonal antibody on promoting NK cells against multiple myeloma cells" have improved in the current revision only in the text part however, the figure quality and labelling of axis in FACS data is still poor. The results presented in the manuscript are quite novel and interesting and therefore, I highly recommend the authors to improve the figure quality.

Especially figures such as Fig 1a, Fig 2e,f are still of sub-optimal figure quality. The correct labeling of Y-axis is important for the figures. For example, Fig2b, and Fig 2i, the Y-axis is incorrectly labeled with the inhibitor name rather than the FACS channel. 

Also the scale bars for all the FACS are missing and important for the final version of the manuscript. I believe the manuscript has merit and can be considered for publication after these figure quality improvements.

Author Response

Point: Especially figures such as Fig 1a, Fig 2e,f are still of sub-optimal figure quality. The correct labeling of Y-axis is important for the figures. For example, Fig2b, and Fig 2i, the Y-axis is incorrectly labeled with the inhibitor name rather than the FACS channel. Also the scale bars for all the FACS are missing and important for the final version of the manuscript. I believe the manuscript has merit and can be considered for publication after these figure quality improvements.

Response: Thank you for your valuable comments. We have marked all the Y-axes as FACS channels. We have reprocessed all the exported flow charts using Adobe Illustrator to standardize the format and save them as vector charts. The scale of the axes has also been added to increase readability.

Round 3

Reviewer 2 Report

The authors addressed all the comments